# Viability Criteria during Liver Ex-Situ Normothermic and Hypothermic Perfusion

**DOI:** 10.3390/medicina58101434

**Published:** 2022-10-11

**Authors:** Fabio Melandro, Riccardo De Carlis, Francesco Torri, Andrea Lauterio, Paolo De Simone, Luciano De Carlis, Davide Ghinolfi

**Affiliations:** 1Hepatobiliary Surgery and Liver Transplantation, University of Pisa Medical School Hospital, Via Paradisa 2, 56124 Pisa, Italy; 2Department of General Surgery and Transplantation, ASST Grande Ospedale Metropolitano Niguarda, 20162 Milan, Italy; 3Department of Surgical, Medical, Biochemical Pathology and Intensive Care, University of Pisa, 56124 Pisa, Italy; 4Department of Medicine and Surgery, University of Milano-Bicocca, 56124 Milan, Italy

**Keywords:** liver transplantation, viability criteria, machine perfusion, hypothermic machine perfusion, normothermic machine perfusion

## Abstract

With the increased use of extended-criteria donors, machine perfusion became a beneficial alternative to cold storage in preservation strategy for donor livers with the intent to expand donor pool. Both normothermic and hypothermic approach achieved good results in terms of mid- and long-term outcome in liver transplantation. Many markers and molecules have been proposed for the assessment of liver, but no definitive criteria for graft viability have been validated in large clinical trials and key parameters during perfusion still require optimization.In this review, we address the current literature of viability criteria during normothermic and hypothermic machine perfusion and discuss about future steps and evolution of these technologies.

## 1. Introduction 

The ongoing discrepancy between organ demand and supply has moved the spotlight toward implementing rescue strategies for extending the pool to deceased donors that were previously considered marginal or even unsuitable for transplantation [1,2,3]. In the last years, ex-situ machine perfusion (MP) emerged as an alternative to static cold storage (SCS) preservation, especially in extended criteria donors (ECD), which have a higher susceptibility to ischemia reperfusion injury (IRI) compared to standard donors [4]. Both normothermic (NMP) and hypothermic machine perfusion (HMP) showed good results in terms of mid- and long-term outcome in liver transplantation (LT), but their role in organ preservation has not been fully elucidated. The decision on which kind of technology, temperature and for how long to preserve liver grafts in MP is still under debate and each transplant center developed its own operative protocols. If the potentiality of MP to minimize post-LT complication is emerging in clinical practice, their capacity to allow a proper organ selection is questionable but represents a fundamental step to increase the pool of available grafts. Many markers and molecules have been proposed for the assessment of the livers (Table 1), but no definitive criteria for graft viability have been validated in large clinical trials and key parameters during perfusion still require optimization. Even if the capacity of NMP to predict organ viability seems more promising, HMP showed great potentialities as well. 

In this review, we evaluate the current literature on MP with the aim to provide a concise overview on the adopted viability criteria during NMP and HMP and discuss about the future steps and the evolution of these technologies.

## 2. Paper Selection

A systematic of the published literature, with the goal to investigate viability criteria in machine perfusion for LT was carried out on the 1 August 2022. Inclusion criteria for this review were as follows: A search of the MEDLINE, Scopus and Cochrane Database was conducted using the following terms: for NMP section: (“normothermic” AND/OR “machine perfusion”) AND liver transplantation” AND ((“2005/01/01”[Date—Publication]: “2022/08/01”[Date—Publication])), for HMP section: ((hypothermic) AND (machine perfusion)) AND ((“2005/01/01”[Date—Publication]: “2022/08/01”[Date—Publication])). The references of each of the selected articles were also evaluated in order to locate additional studies that were not included in the initial search. Only clinical studies on human graft were considered.

The search streategy was performed by the Preferred Reporting Items for Systemic Reviews and Meta-Analysis (PRISMA) guidelines [5] (Figure 1). 

Relevant articles were extracted independently by four authors (F.M., R.D.C, A.L., F.T.) who evaluated and excluded duplicates. No specific search dates were used. Consensus for the relevance of an included study were carried out by four senior authors (R.D.C. and A.L: for HMP chapter; F.M. and D.G. for NMP chapter). Given the heterogeneity of the selected studies and paucity of patients identified within the selection criteria, the results are reported as a narrative review.

## 3. The Evolution of Normothermic Machine Perfusion 

NMP provides a near-physiological environment to the liver, has the potential to evaluate high-risk grafts viability, allow organ therapeutics and improve transplant logistic by prolonging the preservation time up to 24 h [6].

The first phase-1 non-randomized prospective clinical trial evaluating NMP in LT was performed in 2016 by Ravikumar et al. [7] in which 20 liver grafts were preserved with the OrganOx Metra^®^ device for a median of 9.3 h (3.5–18.5h). The study demonstrated that NMP was safe and feasible reporting 100% graft survival at 1 and 6 months in the NMP groups versus 97.5% at the same time points in the static cold storage (SCS) group, respectively. Selzner et al. [8] retrospectively compared SCS to NMP using the albumin-based Steen Solution^TM^ as an alternative to a red blood cell-based perfusate. They reported a 100% 3-months graft survival in the study group and demonstrated the safety of the Steen Solution^TM^ for ex-situ NMP. Bral et al. [9] reported the first NMP experience in Canada in ECD and donation after cardio-circulatory death (DCD) donors. The authors reported comparable results to SCS preserved donors despite more prolonged intensive care and hospital stays in the NMP group. The OrganOx Metra^®^ device was initiated at the donor center. Nasralla et al. [10] published the first prospective randomized controlled trial (RCT) comparing NMP with SCS and showed a 50% lower rate of graft discard, but no significant advantages in terms of graft and patient survival. In the same year, Watson et al. [11] analysed the perfusion characteristics of 47 livers, 22 of which were eventually transplanted, and argued that liver viability during NMP can be assessed using a combination of transaminase release, glucose metabolism, lactate clearance, and maintenance of acid-base balance. Bile pH measurement had a relevant role in prediction of post-transplant ischemic cholangiopathy (IC). The publication of Ghinolfi et al. [12] investigated the efficacy of NMP in very old grafts in a prospective RCT, evaluating graft and patient survival at 6 months post-LT as the primary outcome. No cases of primary non function (PNF) was reported in the NMP group, but the use of NMP did not show any advantages in terms of post-operative transaminases peak, vascular or biliary complications, and length of hospital stays compared to SCS. In order to evaluate the potentiality of NMP to increase the number of available organs, Mergental et al. [13] performed NMP in discarded livers in the so-called VITTAL study. They evaluated 31 grafts and eventually transplanted 22 (71%) of them with a 100% 3-months graft and patient survival. Despite these relevant results, NMP was not able to prevent biliary complications. More recently, a RCT by Markman et al. [14] showed better results in liver perfused by NMP if compared to SCS in terms of early allograft disfunction (EAD) (18% vs. 31%, *p* = 0.01), evidence of IRI at histology (6% vs. 13%, *p* = 0.004), incidence of IC at 6 (1.3% vs. 8.5%, *p* = 0.02) and 12 months after LT (2.6% vs. 9.9%, *p* = 0.02), larger use of DCD livers (51% vs. 26%, *p* = 0.007). Reiling et al. [15] described the first Australian experience with a “back-to-base” NMP approach in rejected livers due to poor graft quality. All 10 cases experienced good outcomes, with no biliary complications, despite 50% of EAD. Similar results are showed in an American experience with discarded grafts reported by Quintini et al. [16] on 21 livers assessed with NMP. In this series one patient developed an IC amenable to endoscopic treatment 4 months after LT. Most important, the authors proposed the normalization of viability criteria per liver weight and perfusate volume. Cardini et al. [17] presented data of prolonged NMP (up to 28 h) routinely used in marginal organs, logistic challenges, and complex recipients. No significant complications were reported, and night-time or parallel procedures were avoided. Eshmuminov et al. [18] developed an own made machine perfusion device with the aim to evaluate very prolonged perfusions of the liver. Ten discarded human grafts underwent NMP for 7 days and 6 out of 10 showed preserved bile production, synthesized coagulation factors and restored cellular energy levels. These results demonstrated that long-term NMP is feasible with the appropriate technology. Finally, the group in Groeningen [19] tested the viability of high-risk livers sequential HMP and NMP. In this prospective clinical trial, the 11 livers that met hepatocellular and cholangiocellular viability criteria were transplanted showing 100% patient and graft survival at 3 and 6 months.

Many metabolic and dynamic parameters are used during ex-situ NMP to evaluate graft quality and viability, but the predictive value of perfusate biomarkers on post-LT outcomes remains to be established. The studies that focused on the liver viability parameters during NMP are summarized in Table 2.

## 4. The Evolution of Hypothermic Machine Perfusion

HMP is gaining increasing widespread acceptance in the preservation of marginal liver grafts [20,21,22,23,24]. Dual hypothermic oxygenated perfusion (D-HOPE) and HOPE have been proven effective in improving the post-transplant outcome of DCD and brain death (DBD) livers in two recent randomized controlled trials [25,26]. The first international study analyzed the role of HMP in DCD liver donors [27] compared to matched SCS DCD livers. HOPE significantly decreased alanine-aminotransferase (ALT) peak (1239 vs. 2065 U/L, *p* = 0.02), IC (0% vs. 22%, *p* = 0.015), biliary complications (20% vs 46%, *p* = 0.042), and 1-year graft survival (90% vs. 69%, *p* = 0.035). Re-transplantation rate was higher in SCS group (18% vs. 0%). Similarly, Guarrera et al. [28] perfused with HMP 31 ECD “orphan livers” which were compared with 30 SCS grafts. Biliary complications (13% vs. 43%) and hospital stay were reduced in HMP group compared with SCS group. In 2018, Schlegel et al. [29] compared 50 DCD livers perfused by HMP with 50 DCD and 50 DBD preserved by SCS. Results showed less IC (8 vs. 22%), acute rejection (4 vs. 28%) and better 5-year patient survival (94% vs. 78%) in HMP group. The RCT led by the Groningen group demonstrated a significant reduction of the incidence and severity of IC in DCD livers preserved with end-ischemic D-HOPE compared to the SCS controls [25]. In another RCT on HOPE in DBD published by the group of Berlin, the HOPE group reported significantly lower transaminase peaks and 90-day complications [26]. Moreover, a recent European series of DBD and DCD liver grafts have shown that (D-)HOPE may be used to safely extend the preservation time and ease transplant logistic [30].

## 5. Viability Parameters during NMP

NMP is an ex-situ technology that maintains the liver at 37 °C in a physiological state through the delivery of oxygen and nutrition. Throughout perfusion, hepatic artery (HA) pressure is set to 70 mm Hg and portal pressure to 6–8 mm Hg. The flow rates targets are >150 mL/minute in the HA and 600–1200 mL/minute in the portal vein (PV). During perfusion, serial arterial perfusate and bile samples are collected and biopsies for serial histological analysis can be obtained. Many metabolic and dynamic parameters are used during ex-situ NMP to evaluate graft quality and viability, but the predictive value of perfusate biomarkers on post-LT outcomes remains to be established. 


**Lactates**


Lactate clearance is the most used viability criteria during NMP evaluation. Lactate is a product of anaerobic glycolysis. The anoxia in the donor is the common cause of lactate elevation. The liver is the major organ responsible for lactate clearance, thus lactate represents a dynamic biomarker to monitor the function of the liver grafts [31]. In the setting of NMP a certain amount of lactates originates from erythrocytes in the perfusate.

In the first experience of Mergental et al. [32] lactate clearance ≤ 2.5 mmol/L achieved within the first 3 h of NMP in combination with evidence of bile production, good liver appearance, glucose metabolism, stable artery flow > 150 mL/min, portal flow > 500 mL/minute, and perfusate pH > 7.3 were used as liver acceptance criteria. The same parameters were considered in the VITTAL trial [13] as well, but the assessment period was prolonged up to 4 h. Lactate clearance was not predictive of IC. Despite lactate is the main parameter in liver acceptance decision, the predictive value during NMP is not defined. Nasralla et al. [10] reported one case of PNF with acceptable lactate clearance during NMP, and Watson et al. [11] reported that the only liver that experienced PNF showed a perfusate lactate level of 2.5 mmol/L after 90 min of perfusion.

Ghinolfi et al. [12,33] proposed a lactate downtrend irrespective to the baseline and final values as marker of liver viability both in old DBD and DCD donors with prolonged warm ischemia time, while Reiling et al. [15] applied a threshold of ≤2.0 mmol/L within the first 2 h of NMP and Van Leeuwen et al. [27] a cut-off ≤1.7 mmol/L. In 2019, Ceresa et al. [34] discarded two livers with lactate ≥ 4 mmol/L after 4 h of perfusion. The next year, Cardini et al. [17] presented 25 successfully LT perfused with NMP running up to 38 h in which a physiological lactate level and a physiological pH value 2 h after commencing perfusion were considered parameters of liver viability. As shown, the timepoint when to decide if accepting a liver based on lactate value is still under debate. Recently Hann et al. [35] showed good outcomes with liver that slowly cleared lactates reaching the level of 2.5 mmol/L only after 6 h.

Quintini et al. [16] argued that perfusate composition and volume, and the graft size are pivotal parameters to be considered in evaluating lactate clearance. Therefore, they proposed the standardized lactic acid clearance (SLAC), which is adjusted by graft size and the amount of circulating perfusate.

To date, not only lactate clearance but a combination with other parameters and the normalization of the lactate values per liver size and amount of perfusate are key factors for the decision to transplant a graft. 


**Transaminases**


Although transaminases value is one of the most used markers of injury during perfusion, its correlation with post-operative outcome is poor and the level during perfusion could be influenced by the “wash-out” phenomenon and the size of the liver [36]. Wash-out depends on amount of preservation fluid resulting in altered lower levels. For these reasons, Quintini et al. [16] recently proposed to normalize transaminases per liver weight and preservation solution volume. Moreover, aspartate aminotransferase (AST) level could be impaired by hemolysis in perfusate, whereas ALT is a more specific liver enzyme. 

In the clinical setting, no defined cut-off values were adopted. Only grafts with very high levels of perfusate transaminases are discarded (ranging from >5000 up to 9000 IU/L based on center experience and preferences). Watson et al. [11] reported a case of PNF in a patient transplanted with a liver with a perfusate ALT level exceeding 9000 IU/L. Similarly, in the Bral et al. [9] series, a liver with perfusate ALT levels > 9000 IU/L require re-LT after 3 months.

Nasralla et al. [10] showed that a higher baseline perfusate ALT level was associated to worst outcome, while Ghinolfi et al. [12] couldn’t find any association between perfusate peak AST level (219 to 3125 IU/L) and post-LT transaminases (*p =* 0.092; *r* = 0.560).


**Glucose Metabolism**


Glucose is also an easy and rapid marker of viability. Initially, glucose in the perfusate is high due to the glycogenolysis activated during SCS. One hour after commencing NMP, functioning livers determine the glucose concentration fall due a block of glycogenolysis and trigger of glycogenesis. Low levels of glucose at NMP start are related to PNF. The stimulation test with exogenous glucose was suggested by Watson [11]. In case of viable liver, the glucose in perfusate rose after administration and subsequently decrease.

Several authors considered glucose metabolism as an important viability marker in multiparametric assessment [11,19,34]. Notably, Mergental criteria [13] include the evaluation of glucose metabolism after 4 h of perfusion together with other viability parameters. 


**Ph**


Perfusate pH is usually low at the beginning of NMP due to hypoxia and anaerobic metabolism. During perfusion, pH must maintain within a physiological range. Bicarbonate may be administered at the beginning to correct pH, but several authors [11] reported unfavourable outcome in liver requiring more than 30 mmol/L of bicarbonate during NMP. Many factors, as perfusate composition, additives, or partial pressures of carbon dioxide, could influence pH. For this reason, pH could be considered as a viability marker only in multiparametric assessment. 


**Platelet and Coagulation Factors**


Production of coagulation factors could show the efficient synthetic function of the liver. Eshmuminov et al. [18] in their study on prolonged NMP, presented data on perfusate factor V concentration which was significantly higher in viable livers after 2 days of perfusion. 

More recently, Weissenbacher et al. [37] tested the capacity of liver synthesis by measuring platelet counts, fibrinogen antigen, von Willebrand factor (vWF) antigen, and coagulation factor XIII-A subunit (FXIII-A) antigen. The authors found that the increased platelets and vWF antigen in the perfusate were predictive of EAD.


**Bile Evaluation**


Bile storage and analysis are routinely performed during NMP with the aim to find markers predictive of IC or biliary complications, which remain the Achille’s heel of LT [38,39,40,41,42,43]. Both cholangiocellular and hepatocellular criteria are relevant in the prediction of post-LT biliary complications. Several studies focused on the evaluation of biliary biomarkers during NMP, mainly pH, glucose, bicarbonate, and electrolytes concentration. 

Watson et al. [11] were the first to postulate that a bile pH > 7.4 was associated to a high risk of developing IC. The Groningen group considered biliary pH as viability criteria for LT, setting a bile pH threshold of >7.48 [42] or >7.45 at 2.5 h of perfusion [23].

Several groups proposed to evaluate the bile in relation to other perfusate parameters. Matton et al. [44] showed that a biliary/perfusate glucose ratio < 0.67, biliary lactate dehydrogenase (LDH) <3689 U/L and biliary bicarbonate < 18 mmol/L are related to high biliary injuries rate. Van Leeuwen et al. [19] proposed the use of the difference between perfusate and bile pH, glucose, and bicarbonate as markers of biliary viability, while Watson et al. [10] showed that a difference between perfusate and bile glucose concentration < 10 mmol/L was associated to a significant injury. Melandro et al. [45] showed biliary good outcome in DBD and DCD exceeding Matton criteria (one case of biliary complications out of 19).

Novel biliary biomarkers have been recently reported in literature. Matton et al. [46] investigated miRNA levels in perfusate and bile during NMP of 12 declined human liver grafts and discovered that cholangiocyte-derived microRNAs (CDmiRNA-222) correlate with biliary injury and function as reflected by LDH, bilirubin and bicarbonate levels. Liu et al. [47] investigated biliary regeneration during 24 h of NMP in 10 discarded livers. The authors showed regeneration of cholangiocytes and peri-biliary glands (PBG) during NMP of steatotic livers as indicated by increased Ki-67 staining in bile duct biopsies.

## 6. Viability Parameters during HMP

Cold preservation relies on the suppression of the metabolic rate, as most enzymatic reactions slow down with temperature reduction. Therefore, hypothermic technologies are traditionally considered less useful in assessing graft viability. Moreover, in the cold, there is a lack of active secretion of bile, which is used as viability parameter during NMP. Even when the perfusion is through the portal vein only, a certain fluid secretion through the biliary tree has been observed during HOPE, corresponding to perfusate mixed with molecules released from the hepatocytes [20]. However, this biliary fluid appears rather unphysiological due to the reduced secretory processes and has not been systematically assessed yet [19,20,21,22,23,24,25,26,27,28]. Therefore, some groups have explored the possibility of combining (D-)HOPE with subsequent controlled oxygenated rewarming and NMP for viability testing [15,48]

According to recent research, the key mechanism of (D-)HOPE seems to be the modification of the mitochondrial metabolism, as reported in mammalian hibernation and suspended animation [21]. The delivery of oxygen under hypothermic conditions induces a slow electron flow through the respiratory chain complexes, thus allowing the recovery of adenosine triphosphate and the metabolism of succinate, ultimately preventing the release of reactive oxygen species (ROS) after warm reperfusion [49]. Therefore, during (D-)HOPE the liver is far from being metabolically inactive, and numerous molecules measured during NMP can be also identified through mass spectrometry from perfusates obtained during (D-)HOPE [20]. The main proposed criteria for liver selection during HMP have been summarized in Table 3.


**Pressure, Flow, and Resistance**


Vascular resistance during HMP is an independent predictor of functional recovery and graft survival in the context of kidney transplantation, nevertheless, the predictive accuracy is modest and vascular resistance is discouraged to be used as the sole basis for organ acceptance [50,51]. A similar correlation in the liver has not been observed, probably due to the different microcirculation of the two organs. Only Liu et al. reported a possible correlation between arterial resistance and the previous warm ischemia in a porcine model, but this finding has never been confirmed by other authors in similar studies [52,53]. Therefore, perfusion pressures, flow and resistance are routinely measured during HMP but they are not currently considered for viability testing [21].


**Transaminase, Lactate Dehydrogenase, Glucose, and Lactate**


In the first clinical series published by Guarrera et al. [23] transaminase peak after liver transplant positively correlated with transaminase and LDH levels in perfusate, with however limited impact on the clinical outcome and complications. Among different perfusion parameters, Patrono et al. [54] found that ALT level at 90 min was the best predictor of EAD with a cutoff of 537 IU/L, although this result was not confirmed at multivariate analysis. Moreover, 90 min LDH had the highest correlation with the liver graft assessment following transplantation (L-GrAFT) risk score [55]. Using microdialysate analysis, the same group found that lactate and glucose release in the interstitial fluid (as an effect of anaerobic glycogenolysis) also correlated with EAD [56]. Nevertheless, as the authors had only a very few cases of graft failure, any correlation with graft survival was not possible [57]. A similar correlation with EAD or post-transplant transaminases peaks was also noted in the study of Muller et al., but perfusate transaminase, glucose, and lactate failed to predict graft failure and complications, thus showing a limited clinical predictive value [58].


**Flavin Mononucleotide**


Among the many functions of the liver and the consequent many confounders, Panconesi et al. [20] have pointed out the necessity to focus on the instigators of IRI, rather than the consequences. The key mechanism of IRI is the production of ROS at mitochondrial complex I, with consequent inflammation. Complex I catalyzed the first step of the mitochondrial electron transfer in the respiratory chain. Metabolomic perfusate analysis has identified a specific fragment of complex I, namely flavin mononucleotide (FMN), as a potential marker of mitochondrial function and injury. Under physiologic conditions, FMN is tightly bound to a specific pocket in complex I but can dissociate from it when complex I is completely reduced, as happens during ischemia [29]. Release of FMN has been reported in cerebral and cardiac mitochondria after transient ischemia [29,59].

The Zurich group has recently found that FMN, determined by fluorescence spectroscopy in HOPE perfusate, correlated with early graft loss, cumulative complications, and hospital stay after liver transplant [29,60]. Based on these observations, if the FMN concentration climbs above 8800 AU at 30 min of HOPE or a sharp incline is seen, the authors recommend not to transplant the liver; while for an intermediate release (8800–5000 AU), they suggest allocating the liver to a low-risk recipient [20,21,22,23,24,25,26,27,28,29]. Some other groups have confirmed the correlation of FMN with posttransplant liver function, and external validation of this method is currently ongoing [20,60]. Nevertheless, its use is still little widespread so far, even among centers that routinely use (D-)HOPE [24].

## 7. Future Perspectives and Conclusions 

Despite the validation in several cohorts, the establishment of reliable markers during NMP and HMP requires higher caseloads and RCT. Nevertheless, these trials can be hardly planned for many reasons: (1) no clear endpoints and graft risk scores are defined in the field of LT and machine perfusion, (2) the presence of several perfusion devices and protocols make multicentric study difficult to be planned, (3) the prohibitive costs discourage the broad utilization of machines [36].

In this scenario, it is very difficult to validate viability parameters during NMP, and lactates and transaminases, the most used markers, have been recently downgraded in several experiences on liver perfusion due to the poor predictive value. 

Graft viability testing and selection can also be performed during (D-)HOPE. Different parameters have been explored, but only FMN correlated with graft failure. Besides graft selection, the addition of specific molecules to limit IRI may be considered in future research [21].

The challenge for the future is to find dynamic, very specific, rapidly measurable markers able to predict postoperative complications and long-term outcome. Technical advance in laboratory technique [61], genomics markers [62,63], new technologies such as hyperspectral real time imaging [64] and artificial intelligence might improve pre-transplant viability assessment and expand the donor pool.

## Figures and Tables

**Figure 1 medicina-58-01434-f001:**
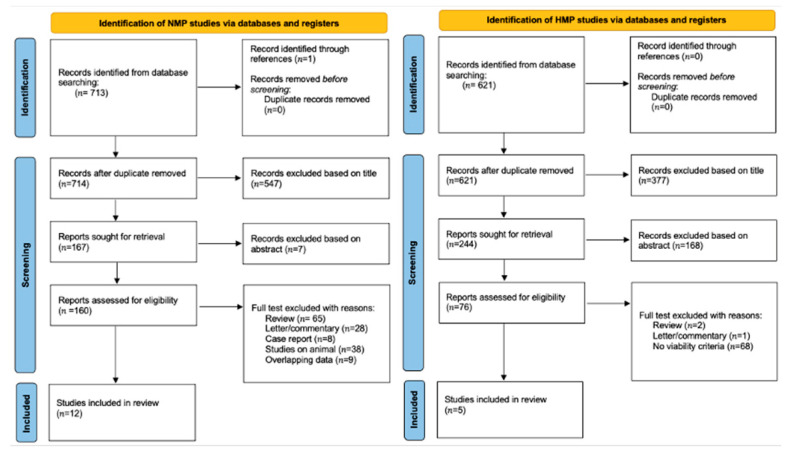
Prisma.

**Table 1 medicina-58-01434-t001:** Overview of viability assessment in ex-situ machine perfusion.

Normothermic Machine Perfusion		
**Blood**	**Hemogasanalysis**	Lactates
		pH
		Glucose Metabolism
		Bicarbonates
	Biochemical analysis	Transaminases
		Platelet and coagulation factors
	Macroscopic appearance	
	Arterial and portal flow	
Bile	Biochemical analysis	Biliary pH
		Biliary bicarbonate
		Biliary glucose
	Biliary production	
**Hypothermic machine perfusion**		
Perfusate	Hemogasanalysis	Glucose, lactate
	Biochemical analysis	AST, ALT, LDH
	Fluorescence spectroscopy	Flavin mononucleotide

**Table 2 medicina-58-01434-t002:** Proposed viability criteria during **Normothermic Machine Perfusion** in clinical studies.

Author, Year, Ref	Country	*n*	Viability Parameter	End-Point	Threshold
**Ravikumar et al., 2016** [7]	UK	20	Perfusate pH, bile production	30-day graft survival	Stable arterial and portal flow; pH beetwen 7.2 and 7.4 without correction; bile production
**Mergental et al., 2016** [32]	UK	6	Perfusate lactate, pH, glucose metabolism, bile production	ITU stay, in-hospital stay	Within 3 h of NMP: Lactate clearance to <2.5 mmol/L or evidence of bile production combined with at least two of the following criteria: 1. Perfusate pH > 7.30 2. Hepatic artery flow >150 mL/min and portal vein flow > 500 mL/min 3. Homogenous perfusion with soft parenchyma consistency
**Bral et al., 2017** [9]	Canada	9	Perfusate lactate, pH, transaminases, bilirubin Bile production	Primary: 30-day graft survival Secondary: Patient survival at day 30, peak serum transaminase AST in first 7 days, EAD incidence in first 7 days, liver biochemistry in serum on days 1–7, 10, and 30, major complications defined by Clavien-Dindo score ≥3, patient and graft survival at 6 months, biliary complications at 6 months	pH, Lactate, ALT, AST, bilirubin, perfusion vascular stability, hourly bile production
**Watson et al., 2018** [11]	UK	47	Perfusate lactate, pH, transaminases, glucose metabolism Bile production, bile pH, bile glucose	PNF, EAD, biliary complications	1. Peak lactate fall ≥ 4.4 mmol/L/kg/h 2. ALT<6000iU/Lat2h 3. Maximum bile pH > 7.5 4. Bile glucose ≤ 3 mmol/L or 10 mmol less than perfusate glucose 5. Maintain perfusate pH > 7.2 with ≤30 mmol bicarbonate supplementation 6. Falling glucose beyond 2 h OR perfusate glucose < 10 mmol/L with subsequent fall during challenge with 2.5 g glucose
**Ghinolfi et al., 2018** [12]	Italy	10	Perfusate lactate	Graft and patient survival at 6 months	1. Lactate downtrend 2.S table flow 3. Acceptable gross appearance with uniform vascularization.
**Van Leeuwen et al., 2019** [19]	Netherland	16	Perfusate lactate, pH, Bile production, bile pH	Primary: 3-month graft survival Secondary: graft and patient survival at 6 months, PNF, biliary complications, biochemical serum markers of graft function and ischemia–reperfusion injury at postoperative days 1 to 7, and after 1 and 3 months; graft utilization rate.	After 2.5 h of NMP: 1. Lactate clearance to ≤1.7 mmol/L. 2. Perfusate pH 7.35–7.45. 3. Bile production > 10 mL. 4. Biliary pH > 7.45
**Matton et al., 2019** [44]	Netherland	23	Perfusate lactate, pH, Bile production, bile pH	Biliary complications	After 2.5 h of NMP: 1. Lactate clearance to ≤1.7 mmol/L; 2. Perfusate pH 7.35–7.45; 3. Bile production > 10 mL; 4. Biliary pH > 7.48
**Mergental et al., 2020** [13]	UK	31	Perfusate lactate, pH, glucose metabolism, bile production	90-day patient and graft survival	Within 4 h of NMP: lactate < 2.5 mmol/L and ≥2 of the following criteria: 1. Evidence of bile production; 2. pH > 7.30; 3. Metabolism of glucose; 4. HA flow > 150 mL/min and PV flow > 500 mL/min; 5. Homogenous perfusion
**Cardini et al., 2020** [17]	Austria	34	Perfusate lactate, pH, transaminases; Bile production, bile pH	Graft and patient survival, Biliary complications	1. Rapid decrease and maintenance of lactate levels (first 2 h of NMP) 2. Bile output and biliary pH. 3. Maintenance of physiological perfusate pH without sodium bicarbonate. 4. Exceptionally high or sharp incline of AST, ALT, LDH.
**Reiling et al., 2020** [15]	Australia	10	Perfusate lactate, pH, glucose metabolism Bile production	EAD, graft and patient survival	Within 2 h (to 4 h) of NMP: 1. Lactate clearance to <2 mmol/L 2. Decreasing trend in perfusate glucose concentration by 4 h. 3. Physiological pH without the need for sodium bicarbonate. 4. Stable HA and PV flows. 5. Homogeneous graft perfusion with soft parenchyma consistency 6. Evidence of bile production
**Markmann et al., 2020** [14]	USA	153	Perfusate lactate	incidence of EAD	Perfusate lactate
**Quintini et al., 2022** [16]	USA	21	Perfusate lactate (standardized lactic acid clearance [SLAC]) measurements by converting the lactate concentration to the percentage of lactate clearance per liver weight at each time point during perfusion), bile production	EAD, PNF, biliary complications	Within 6 h of NMP, at least 2 of the following criteria: 1 lowest perfusate lactate level <4.5 mmol/L or a decrease of 60% from peak in the first 4 h.2. bile production rate higher than 2 mL/h; 3. stable HA flow of >0.05 mL/ min/g of liver weight and PV flow >0.4 mL/min/g of liver weight. 4. macroscopic homogenous perfusion and soft consistency.

ALT, alanine aminotransferase; AST, aspartate aminotransferase; ITU: intensive therapy unit; NMP: normothermic machine perfusion; UK, United Kingdom; EAD, early allograft dysfunction; PNF: primary non function; LDH: lactate dehydrogenase; PRS: post reperfusion syndrome; RRT: renal replacement therapy; HA: hepatic artery; US, United States.

**Table 3 medicina-58-01434-t003:** Proposed viability criteria during hypotermic machine perfusion in clinical studies.

Author, Year	Country	n	Type of HMP	Viability Parameter	End-Point	Threshold
**Guarrera et al., 2010** [50]	US	20	HMP (w/o oxygen)	Perfusate AST, ALT, and LDH	Correlation with peak serum AST and ALT posttransplant	N/A
**Muller et al., 2019** [53]	Switzerland	54	HOPE	Perfusate FMN (30 min)	Correlation with 90-day graft loss	10,000 AU
**Patrono et al., 2020** [51]	Italy	50	DHOPE	Perfusate ALT (90 min)	Correlation with EAD	537 IU/L
				Perfusate LDH (90 min)	Correlation with L-GrAFT score	N/A
**Schlegel et al., 2020** [21]	Switzerland	50	HOPE	Perfusate FMN (30 min)	Correlation with graft loss	8000 AU
**Patrono et al., 2022** [52]	Italy	10	DHOPE	Microdialysis glucose and lactate (2 h)	Correlation with EAD and L-GrAFT score	N/A

ALT, alanine aminotransferase; AST, aspartate aminotransferase; AU, arbitrary units; (D)HOPE, (dual) hypothermic oxygenated perfusion; EAD, early allograft dysfunction; FMN, flavin mononucleotide; HMP, hypothermic machine perfusion; L-GrAFT, liver graft assessment following transplantation; LDH, Lactate dehydrogenase; US, United States.

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
