# Peer review of "Viability Criteria during Liver Ex-Situ Normothermic and Hypothermic Perfusion"

_medicina, 2022, doi:10.3390/medicina58101434_

Round 1
Reviewer 1 Report
Dear Authors,
The article contains current and valuable information in the field of liver transplantation. In recent years, donor criteria have been expanded in order to increase the donor pool due to organ limitation. The use of donors with cardiac death continues at an increasing rate in most countries. For these reasons, the use of normothermic machine perfusion has become popular in recent years to avoid primary graft dysfunction. Therefore, I think that the scientific contribution of this review is important.
For this reason, thank you for this article
Author Response
Dear Authors,
The article contains current and valuable information in the field of liver transplantation. In recent years, donor criteria have been expanded in order to increase the donor pool due to organ limitation. The use of donors with cardiac death continues at an increasing rate in most countries. For these reasons, the use of normothermic machine perfusion has become popular in recent years to avoid primary graft dysfunction. Therefore, I think that the scientific contribution of this review is important.
For this reason, thank you for this article
A. We thank the reviewer for the fine revision and the constructive comments.

Reviewer 2 Report
The authors provide a comprehensive review of literature on the topic “ex-situ HMP and NMP perfusion of liver grafts and gave an overview about the used biomarkers to assess the viability.
Minor comments:
1. In the title, the type of organ is missing as it could also be NMP or HMP of kidney grafts.
2. The authors should clarify in the very beginning of the story, if they are going to summarize data of human liver grafts only, or if they are going to assess also experimental data, because there is a lot of literature out there dealing with data from animal experiments
Major comments:
1. The 2nd minor comment brings raises the question for why not assessing experimental data on this topic? There is a lot of literature available dealing with new biomarkers, which were promising, e.g. extracellular vesicles, respectively exosomes (PMID: 35243279, PMID: 31545919). In the Reviwer´s eyes, it would be beneficial, to have an additional overview about biomarkers which were used in experimental settings and how the potential for the translation into the clinics is rated.
2. The authors have an overview in table format concerning the relevant studies in this field, however, selecting relevant stories by only two persons, in this case senior authors, carries a large risk of bias. Additionally the search strategy appears wrong to the reviewer, because if performing the same search on pubmed, >120k hits were found, if adding the single search criteria by “or” in the advanced search mask. Doing it with “And” there is a result of zero hits, which makes sense, because HMP and NMP at the same time is not sense full. But this means, the reader is not able to recreate the search strategy which was used here and therefore, it is not possible to judge if all relevant papers were found, and additionally, how and why the selected papers were chosen. Thus, from the methodology, this search is not adequate for a publication and the risk for bias is far too high
3. In the reviewers eyes, the search has to be performed in a new and correct way and interpreted and summarized completely new, and then, additionally to the tables presented in here, the main outcome should be another table summarizing the mainly used biomarker as kind of core set for viability assessment. This is what the title promises to the reader, but the reader has to search the most commonly used markers out of the large summary tables
Author Response
The authors provide a comprehensive review of literature on the topic “ex-situ HMP and NMP perfusion of liver grafts and gave an overview about the used biomarkers to assess the viability.
Minor comments:
- In the title, the type of organ is missing as it could also be NMP or HMP of kidney grafts.
A. Thank you for this suggestion. We changed the tile in “Viability criteria during liver ex-situ normothermic and hypothermic perfusion“
- The authors should clarify in the very beginning of the story, if they are going to summarize data of human liver grafts only, or if they are going to assess also experimental data, because there is a lot of literature out there dealing with data from animal experiments
A. Thank you for this suggestion. In order to improve clarity, We have provided additional text as follows: “Only clinical studies on human graft were considered”
Major comments:
- The 2nd minor comment brings raises the question for why not assessing experimental data on this topic? There is a lot of literature available dealing with new biomarkers, which were promising, e.g. extracellular vesicles, respectively exosomes (PMID: 35243279, PMID: 31545919). In the Reviwer´s eyes, it would be beneficial, to have an additional overview about biomarkers which were used in experimental settings and how the potential for the translation into the clinics is rated.
A. Thank you very much for this valuable comment. The primary scope of this review is offer to physicians a concise overview of clinical viability parameters and a tool to help them during their clinical practice. Nonetheless, we appreciate your suggestion and we added the two references in the chapter “Future perspectives and conclusions”
- The authors have an overview in table format concerning the relevant studies in this field, however, selecting relevant stories by only two persons, in this case senior authors, carries a large risk of bias.
A. Thank you for pointing this out. We apologize for the lack of information. Selection of relevant articles was performed by two authors for the HMP chapter (RDC and AL and two authors for NMP chapter (FM and DG ), to reduce the risk of bias.
Additionally the search strategy appears wrong to the reviewer, because if performing the same search on pubmed, >120k hits were found, if adding the single search criteria by “or” in the advanced search mask. Doing it with “And” there is a result of zero hits, which makes sense, because HMP and NMP at the same time is not sense full. But this means, the reader is not able to recreate the search strategy which was used here and therefore, it is not possible to judge if all relevant papers were found, and additionally, how and why the selected papers were chosen. Thus, from the methodology, this search is not adequate for a publication and the risk for bias is far too high
A. We thank the Reviewer for their comment and the opportunity to clarify this point. As suggested, we have now added the research strings for PubMed:
- For NMP: “("normothermic" AND/OR "machine perfusion") AND liver transplantation”= 712 results from 2005/01/01 to 2022/08/01
- For HMP: “("hypothermic" AND/OR "machine perfusion") AND liver transplantation” = 729 results from 2005/01/01 to 2022/08/01.
Our original intent was to perform a narrative review, not a systematic one. This does not necessarily mean that the methodology we adopted is unsafe. We excluded preclinical and animal studies and only focused on clinical experiences. Provided that viability assessment is the main target of our manuscript, we carefully evaluated papers providing selection criteria during MP and reported them in Table 2 (for NMP) and Table 3 (for HMP). Relevant papers on NMP/HMP which not provided information about viability assessment were cited in the introductive paragraphs.
- In the reviewers eyes, the search has to be performed in a new and correct way and interpreted and summarized completely new, and then, additionally to the tables presented in here, the main outcome should be another table summarizing the mainly used biomarker as kind of core set for viability assessment. This is what the title promises to the reader, but the reader has to search the most commonly used markers out of the large summary tables
A. Thank you. We apologize for lack of information. As previously reported, we added in the text the research strategy and we wanted to explain the paper’ philosophy. With the intent to better explain our intents, we added in the introduction section a new table summarizing the mainly used biomarkers.

Reviewer 3 Report
Overall well-written, albeit low yield, review article highlighting the current challenges in viability assessment during machine perfusion of livers for transplantation. Some minor grammar editing throughout would be of benefit but otherwise the literature is nicely summarized and accompanying tables are helpful. No significant/valuable data or theories are contributed compared to prior published review papers on the subject.
Author Response
Overall well-written, albeit low yield, review article highlighting the current challenges in viability assessment during machine perfusion of livers for transplantation. Some minor grammar editing throughout would be of benefit but otherwise the literature is nicely summarized and accompanying tables are helpful. No significant/valuable data or theories are contributed compared to prior published review papers on the subject.
A. Thank you very much for your valuable comments. We entirely agree with reviewer’s concerns but it should be considered that several new important papers reporting viability criteria in HMP and NMP (e.g. Quintini et al., 2022) have been published after the main reviews on the field namely Panconesi et al., Brüggenwirth et al.

Round 2
Reviewer 2 Report
The authors add the search strytegy as recommended by the reviewer, however the authors should include a kind of flow chart indicating the results of the search, as given in the rebulttal letter:
- For NMP: “("normothermic" AND/OR "machine perfusion") AND liver transplantation”= 712 results from 2005/01/01 to 2022/08/01
- For HMP: “("hypothermic" AND/OR "machine perfusion") AND liver transplantation” = 729 results from 2005/01/01 to 2022/08/01.
Additionally the number of the finally included references due to the selection process "only focussed on clinical experience" must be shown in this flow chart, this would clarify to the reader, that out of 1441 initial hits, 11? references were chosen for table 2.
Thus, the reader would be able to judge if this search and selection process is adequate and the included hits are representative.
This is teh main outcome of the search, this must be shown, otherwise the overall context is not clear
Author Response
The authors add the search strytegy as recommended by the reviewer, however the authors should include a kind of flow chart indicating the results of the search, as given in the rebulttal letter:
- For NMP: “("normothermic" AND/OR "machine perfusion") AND liver transplantation”= 712 results from 2005/01/01 to 2022/08/01
- For HMP: “("hypothermic" AND/OR "machine perfusion") AND liver transplantation” = 729 results from 2005/01/01 to 2022/08/01.
Additionally the number of the finally included references due to the selection process "only focussed on clinical experience" must be shown in this flow chart, this would clarify to the reader, that out of 1441 initial hits, 11? references were chosen for table 2.
Thus, the reader would be able to judge if this search and selection process is adequate and the included hits are representative.
A. Thank you for the suggestion.
The text has been amended accordingly based on reviewer’s recommendation and the flowchart (PRISMA) was added to the text. The sentence "Only clinical studies on human graft were considered" was added to the text. In the flowchart we highlighted that animal studies are not considered. We thank the reviewers for their thorough analysis and appreciate the opportunity to enhance the manuscript. We hope that the manuscript is now acceptable for publication. If you have any questions, please do not hesitate to contact us.Sincerely,
Fabio Melandro